# Transcriptional and Metabolic Response of a Strain of *Escherichia coli* PTS^−^ to a Perturbation of the Energetic Level by Modification of [ATP]/[ADP] Ratio

**DOI:** 10.3390/biotech13020010

**Published:** 2024-04-10

**Authors:** Sandra Soria, Ofelia E. Carreón-Rodríguez, Ramón de Anda, Noemí Flores, Adelfo Escalante, Francisco Bolívar

**Affiliations:** 1Departamento de Ingeniería Celular y Biocatálisis, Instituto de Biotecnología, Universidad Nacional Autónoma de México, Cuernavaca 62210, Mexico; lasobiotc@gmail.com (S.S.); efelio2000@gmail.com (O.E.C.-R.); daramon.h@gmail.com (R.d.A.); noemi.flores@ibt.unam.mx (N.F.); 2Laboratorio de Soluciones Biotecnológicas (LasoBiotc), Montevideo 11800, Uruguay

**Keywords:** *Escherichia coli* PTS^−^, F_0_-F_1_ ATPase synthase, [ATP]/[ADP] ratio, central carbon metabolism, transcriptional response

## Abstract

The intracellular [ATP]/[ADP] ratio is crucial for *Escherichia coli*’s cellular functions, impacting transport, phosphorylation, signaling, and stress responses. Overexpression of F_1_-ATPase genes in *E. coli* increases glucose consumption, lowers energy levels, and triggers transcriptional responses in central carbon metabolism genes, particularly glycolytic ones, enhancing carbon flux. In this contribution, we report the impact of the perturbation of the energetic level in a PTS^−^ mutant of *E. coli* by modifying the [ATP]/[ADP] ratio by uncoupling the cytoplasmic activity of the F_1_ subunit of the ATP synthase. The disruption of [ATP]/[ADP] ratio in the evolved strain of *E. coli* PB12 (PTS^−^) was achieved by the expression of the *atpAGD* operon encoding the soluble portion of ATP synthase F_1_-ATPase (strain PB12AGD^+^). The analysis of the physiological and metabolic response of the PTS^−^ strain to the ATP disruption was determined using RT–qPCR of 96 genes involved in glucose and acetate transport, glycolysis and gluconeogenesis, pentose phosphate pathway (PPP), TCA cycle and glyoxylate shunt, several anaplerotic, respiratory chain, and fermentative pathways genes, sigma factors, and global regulators. The *apt* mutant exhibited reduced growth despite increased glucose transport due to decreased energy levels. It heightened stress response capabilities under glucose-induced energetic starvation, suggesting that the carbon flux from glycolysis is distributed toward the pentose phosphate and the Entner–Duodoroff pathway with the concomitant. Increase acetate transport, production, and utilization in response to the reduction in the [ATP]/[ADP] ratio. Upregulation of several genes encoding the TCA cycle and the glyoxylate shunt as several respiratory genes indicates increased respiratory capabilities, coupled possibly with increased availability of electron donor compounds from the TCA cycle, as this mutant increased respiratory capability by 240% more than in the PB12. The reduction in the intracellular concentration of cAMP in the *atp* mutant resulted in a reduced number of upregulated genes compared to PB12, suggesting that the mutant remains a robust genetic background despite the severe disruption in its energetic level.

## 1. Introduction

The F_0_-F_1_ ATPase synthase or H^+^-ATPase catalyzes the synthesis of ATP and inorganic phosphate and plays a crucial role in free energy transduction. The synthase is comprised of two complexes or subunits, F_0_ and F_1_. The inner membrane-embedded F_0_ complex consists of the three catalytic subunits a, b, and c (1:2:10), and forms a proton channel through the membrane where proton translocation occurs. The catalytic and cytoplasmic F_1_ complex consists of the α, β, γ, δ, and ε subunits (3:3:1:1:1). Central and peripheric stalks link F_0_ and F_1_ subunits together. The central stalk and a ring of the subunit c of the F_0_ complex form a rotating structure whose rotation is driven by extracellular proton translocation across the F_0_ complex, resulting in structural changes in the F_1_ complex, leading in the synthesis of ATP from ADP and phosphate [1,2,3,4,5]. ATP plays a fundamental role in the cell’s physiology by providing energy for essential processes such as active transport, central carbon metabolism, biosynthetic pathways, DNA, RNA, protein synthesis, signal transduction, motility, cellular division, and stress responses [6]. Under aerobic conditions, the H^+^-ATPase catalyzes the phosphorylation of ADP to ATP by using the proton motive force. Under anaerobic conditions, it energizes the inner membrane by proton extrusion dependent on ATP hydrolysis [2,4,7].

The ATP synthase operon consists of nine genes *atpIBEFHAGDC.* The *atpBEF* genes encode the F_0_ subunits a, b, and c, respectively, whereas the *atpHAGDC* encodes the F_1_ subunits δ, α, γ, β, and ε. The *atpI* gene encodes for the accessory factor AtpI. The entire operon is transcribed as a polycistronic unit [2,4]. The inactivation or overexpression of the partial or entire *atp* operon results in perturbed ATP levels by modification of the [ATP]/[ADP] ratio, affecting the cell’s metabolism and physiology significantly [8].

The expression of the F_1_ subunit of the ATP synthase in the strain of *E. coli* BOE270 by plasmid-cloning the genes *atpAGD* (encoding the α, γ, and β subunits) resulted in a cytoplasmic uncoupled activity, affecting the energy state of the cell, reflected by a lower [ATP]/[ADP] ratio and increased carbon flux through the glycolytic pathway (up to 70%) associated with a smaller decrease in the growth rate in the resultant derivative versus the parental strain. These results proposed that the control of the glycolysis resides outside the pathway, particularly on the enzymes hydrolyzing ATP [9]. Overexpression and inactivation, both in the F_1_ subunit of the ATP synthase in *E. coli* JM101, and the culture under aerobic conditions in an M9 mineral broth supplemented with glucose as the carbon source under aerobic conditions, resulted in derivative strains JMAGD^+^ and JM*atp*^−^, respectively, with relevant metabolic changes. Both derivatives showed reduced energetic levels determined with the [ATP]/[ADP] ratio and cAMP level. Transcriptomic analysis of 105 genes, including the glycolytic and pentose phosphate pathways, TCA cycle, respiratory genes, acetate metabolism, and some regulators, showed a downregulation of the glycolytic, TCA cycle, and respiratory genes in both strains compared to the parental strain JM101 in response to a decrease in [ATP]/[ADP] ratio, and was associated with a reduction in the levels of cAMP. Still, in contrast, derivatives showed relevant increases in glucose and oxygen-specific consumption rates, a relevant trait for industrial production strains [6].

*Escherichia coli* PB12 is a Δ*ptsHIcrr* (PTS^−^ mutant) derivative strain of the parental strain JM101 [10]. This PTS^−^ mutant merged from an adaptive laboratory evolution experiment showing an increased growth capability in glucose as the carbon source by selecting the galactose permease (GalP) as an alternative glucose transporter. Remarkably, this derivative showed an increased carbon flux through glycolysis by 21.54% and in some of the TCA cycle reactions, like the malic enzymes (by 4%), compared to the wild strain JM101 [11], associated with a permanent carbon scavenging condition and the upregulation of several genes of the central carbon metabolism, the TCA cycle and the glyoxylate shunt, acetate production and utilization, and some regulatory genes [11]. To understand the capacity of this evolved mutant to adapt to a drastic energetic perturbation, in this contribution, we report the impact of the perturbation of the energetic level in the PTS^−^ mutant modifying the [ATP]/[ADP] ratio by uncoupling the cytoplasmic activity of the F_1_ subunit of the ATP synthase. We analyzed the physiological and metabolic response of the strain PB12 by determining the *q*_Glc_, *q*_Ace_, and *q_O_*_2_ values in response to the reduction of the [ATP]/[ADP] ratio. Additionally, we explored the transcriptomic response of the cell using RT–qPCR to the energetic perturbation with the analysis of 96 genes involved in glucose and acetate transport, glycolysis and gluconeogenesis, PPP, TCA cycle and glyoxylate shunt, several anaplerotic, respiratory chain, and fermentative pathways genes, sigma factors, and several global regulators.

## 2. Materials and Methods

### 2.1. Bacterial Strains, Plasmids, Culture Media, and Growing Conditions

The bacterial strains JM101, PB12, and PB12AGD^+^ used in this contribution are described in Table 1. Plasmids pTrc99A [12] and pTrc*atpAGD* [6] were used to transform by electroporation of competent cells of *E. coli* PB12 by standard procedures [13]. Cultures were performed using quadruplicate, as performed previously [6,11], on 1 L Applikon ADI 1010 fermenters on M9 medium [13] with 2 g/L of glucose as the carbon source with a working volume of 750 mL, pH 7.0 controlled with NH_4_OH 2.9%, at 37 °C, 600 rpm, an airflow rate of 1 vvm. Cultures started with an OD_600nm_ of 0.05. Samples containing 50 mL of the different strains growing logarithmically in the reactor were withdrawn at OD_600nm_ = 1.0 corresponding to the mild-exponential growth phase [6,11] for ATP and ADP concentrations ([ATP]/[ADP] ratio), ATPase activity, cAMP measurements, and for total RNA extraction for RT–qPCR analysis [6].

The energetic level was estimated as reported previously by using the [ATP]/[ADP] ratio based on the method reported by [6,9]. Aliquots of 40 mL of cultures were withdrawn from bioreactors when the cellular growth was an OD_600nm_ = 1.0, corresponding to the mild-exponential growth phase. Cell lysis was performed as described by [9]. According to the supplier specifications, the ATP concentration was determined using the ENLITEN^®^ ATP Assay System (PROMEGA, Madison, WI, USA). For the [ADP] determinations, the ADP content in the same sample used for ATP determination was converted to ATP by adding 1 U of pyruvate kinase enzyme (PK) (Sigma-Aldrich, St. Louis, MO, USA) and 1 mM phosphoenolpyruvate (PEP) and recorded as the increase in luminescence. Finally, a standard ATP concentration (1 μM) was added to quantify [ATP] and [ADP]. The results were corrected for quenching of the ATP signal by the PK preparation. Soluble ATPase activities were measured in enzymatic assays using the above cellular extracts. The intracellular cAMP concentration determinations were measured in bioreactor cultures at OD_600nm_ = 1.0 using Cyclic AMP Enzyme Immunoassay Kit (Enzo Life Sciences, Farmingdale, NY, USA) following the manufacturer’s instructions.

For ATPase activity, withdrawn cells were centrifuged as above and resuspended in 2.0 mL of ATP buffer (0.1 M Tris-acetate, pH 7.75, 10 mM potassium acetate, 2.0 mM EDTA) and supplemented with 10 µM MgCl_2_ per mL. Resuspended cells were lysed and centrifuged as above, and incubated for 5 min at 37 °C. According to the supplier specifications, ATPase activity was determined using the ENLITEN^®^ ATP Assay System (PROMEGA, Madison, WI, USA). The plasmid *atpAGD* encodes the F1 ATP part of the membrane-bound (F_1_-F_0_) H+ ATP synthase. The expression of this operon in the PTS^−^ mutant PB12 results in an uncoupled cytoplasmic ATPase activity. As enzymatic assays for determination of [ATP]/[ADP] ratio were performed in cellular extracts, results in the unique cellular enzymatic activities, and was considered as the total determined ATPase activity.

### 2.2. Determination of Uncoupled ATPase Activity and Determination of [ATP]/[ADP] Ratio and cAMP Concentration

The energetic level was estimated, as reported previously, by using the [ATP]/[ADP] ratio based on the method reported by [6,9]. Aliquots of 40 mL of cultures were withdrawn from bioreactors when the cellular growth was an OD_600nm_ = 1.0, corresponding to the mild-exponential growth phase. Cell lysis was performed as described by [9]. According to the supplier specifications, the ATP concentration was determined using the ENLITEN^®^ ATP Assay System (PROMEGA, Madison, WI). For the [ADP] determinations, the ADP content in the same sample used for ATP determination was converted to ATP by adding 1 U of pyruvate kinase enzyme (PK) (Sigma-Aldrich) and 1 mM phosphoenolpyruvate (PEP) and recorded as the increase in luminescence. Finally, a standard ATP concentration (1 μM) was added to quantify [ATP] and [ADP]. The results were corrected for quenching of the ATP signal by the PK preparation. Soluble ATPase activities were measured in enzymatic assays using the above cellular extracts. The intracellular cAMP concentration determinations were measured in bioreactor cultures at OD_600nm_ = 1.0 using Cyclic AMP Enzyme Immunoassay Kit (Enzo Life Sciences), following the manufacturer’s instructions.

For ATPase activity, withdrawn cells were centrifuged as above, and resuspended in 2.0 mL of ATP buffer (0.1 M Tris-acetate, pH 7.75, 10 mM potassium acetate, 2.0 mM EDTA) and supplemented with 10 µM MgCl_2_ per mL. Resuspended cells were lysed and centrifuged as above, and incubated for 5 min at 37 °C. According to the supplier specifications, ATPase activity was determined using the ENLITEN^®^ ATP Assay System (PROMEGA, Madison, WI). As the plasmid *atpAGD* encodes the F1 ATP part of the membrane-bound (F_1_-F_0_) H^+^ ATP synthase, the expression of this operon in the PTS^−^ mutant PB12 results in an uncoupled cytoplasmic ATPase activity. As enzymatic assays for the determination of [ATP]/[ADP] ratio were performed in cellular extracts, resulting in unique cellular enzymatic activities, and was considered as the total ATPase activity determined [6,9].

### 2.3. Measurement of Glucose and Acetate from Fermentation Supernatants

The specific growth rate (μ, h^−1^), glucose consumption rate (*q_S_*, g _Glc_ g _DWC_ h^−1^), and acetic acid production for analyzed strains were determined as previously [6,11,16]. Residual glucose and acetic acid were measured using high-performance liquid chromatography (HPLC) in a Waters system (Milford, MA, USA) coupled to a 600E pump, an automatic injector 717, a refractive index of 2410, and a diode array detector. The samples were analyzed in an Aminex HPX-87H column (300 × 7.8 mm, 9 m) Bio-Rad (Hercules, CA, USA), using a 5 mM H_2_SO_4_ as mobile phase at a flow rate of 0.5 mL min^−1^ at 50 °C.

### 2.4. Total RNA Extraction and cDNA Synthesis

Total RNA from the strains was isolated and purified using hot phenol equilibrated with water [11]. The total RNA was precipitated with 3M sodium acetate/ethanol and centrifuged at 10,000 rpm for 15 min at 4 °C; the supernatant was discarded, and the total RNA was suspended in free-RNase water and stored at −70 °C. RNA samples were treated with the Ambion™ DNase I (RNase-free) kit (Thermo Scientific, Waltham, MA, USA), and integrity was analyzed in formaldehyde agarose gel. RNA concentrations were determined using a NanoDrop 2000 (Thermo Scientific). Four RNA extractions and purifications were carried out from four independent bioreactors. cDNA was synthesized using the RevertAid H Minus First Strand cDNA Synthesis Kit Revert Aid^TM^ (Thermo Scientific) using a mixture of specific DNA primers designed using the Primer Express 2.0 software (Perkin Elmer/Applied Biosystems, Waltham, MA, USA).

### 2.5. Transcriptomic Analysis Using RT–qPCR

RT–qPCR reactions were performed, as previously, in an ABI Prism7000 Sequence Detection System (Perkin-Elmer/Applied Biosystems) using the Applied Biosystems™ SYBR™ Green SYBR Green PCR Master Mix (Thermo Scientific) [6,10,11]. Amplification conditions were 10 min at 95 °C and a two-step cycle at 95 °C for 15 s and 60 °C for 60 s, for a total of 40 cycles using specific primers for 96 genes (Appendix A) encoding proteins and enzymes of glycolysis and gluconeogenesis, ATP synthase, the tricarboxylic acids cycle (TCA) and glyoxylate shunt, the pentose phosphate pathway (PPP), anaplerotic and fermentative pathways, respiratory chain, glucose and acetate transport, several sigma factors and global regulators in a final primer concentration or 0.2 mM in a total volume of 15 μL, and 5 ng of cDNA. All experiments were performed in triplicate for each gene of each strain, obtaining very similar values with differences of less than 0.3 SD. A non-template control reaction mixture was included for each gene. The quantification technique used to analyze data was the 2^−ΔΔCT^ method [17]. RT–qPCR results were normalized using the *ihfB* gene as an internal control (housekeeping gene) [6,10,11]. The RT–qPCR expression values for each gene differ by less than 30%. A gene was considered upregulated when the relative transcription level was ≥1.7, and down-regulated when it was ≤0.588. The RT–PCR expression values obtained for each gene differ between them in most of the genes by less than 30%.

## 3. Results

### 3.1. Physiological Parameters of Analyzed Strains of E. coli in Bioreactor Cultures

To elucidate the physiological changes in response to the disruption of the [ATP]/[ADP] ratio in the evolved strain of *E. coli* PB12 (PTS^−^) by the expression of the *atpAGD* operon encoding soluble portion of ATP synthase F_1_-ATPase, the strain PB12 was transformed with the plasmid pTrc*atpAGD* (strain PB12AGD^+^) and grown in 1 L fermenter in M9 medium supplemented with 2 g/L of glucose as the carbon source. Growth, glucose consumption, and acetate production were determined and compared with strains PB12 and JM101 (Figure 1). Table 2 shows kinetic parameters determined for the analyzed strains. The derivative PB12AGD^+^ showed a μ = 0.24 h^−1^, reducing the growth by 55.8% compared to the PB12 in response to cloning the *atpAGD* genes. Associated with this decrement in the μ, cultures of the derivative PB12AGD^+^ showed lower growth, reached the stationary phase at 12 h of cultivation, and showed a final biomass ~0.42 gDCW, compared to the strain PB12, with a biomass of ~0.80 gDCW, and entering the stationary phase at 7.5 h of cultivation (Figure 1). The empty plasmid control strain PB12/pTrc99A was closely related μ to the strain PB12.

Regarding glucose consumption, PB12 depleted glucose at 9 h of cultivation with a q_Glc_ (g_Glc_/g_DCW h_) of 0.81 for PB12, associated with a biomass yield on glucose of 0.53 (g/g), respectively. Derivative PB12AGD^+^ consumed all glucose at 10 h of cultivation, showing a higher q_Glc_ (1.42 g_Glc_/g_DCW h_) but a lower biomass yield on glucose = 0.17 g/g (Table 2) than the PB12 (increased *q_Glc_* = 175.3% and a reduction = 67.92 in YX_/Glc_). The highest acetic acid concentration (g/L) was detected at 11 h of cultivation for PB12, accumulating 0.13 g/L. The acetic acid concentration was 0.093 g/L, and was detected at 13 h of cultivation for PB12AGD^+^. *q_Ace_* (g/gDCW h) values for each strain are shown in Table 2.

### 3.2. [ATP]/[ADP] Ratio, ATPase Activity, and cAMP Concentration Determinations

The energetic level of the studied strains was determined using an analysis of the [ATP]/[ADP] ratio, ATPase activity, and cAMP concentration. All determinations were performed in the analyzed strains growing in aerobic bioreactor cultures when DO_600nm_ = 1.0. Results are shown in Table 2. The parental strain JM101 showed the highest [ATP]/[ADP] ratio = 7.32. This result is consistent with previous reports for parental strains of *E. coli* ranging between 6–10 [9] and for the strain JM101, as determined previously [6]. The ratio [ATP]/[ADP] decreased by 65% in the PB12 strain and 84% in the mutant PB12AGD^+^ and compared to JM101, respectively, corresponding to a decrement of 52% in the *apt* mutant than the observed in PB12 (Table 2).

As stated, enzymatic assays were performed from cellular extracts; thus, the ATPase activity encoded with the plasmid-cloned *atpAGD* operon in the PB12 strain is considered as the total ATPase activity as it is not coupled to any other cellular enzymatic activity. The strains JM101 and PB12 showed practically the same ATPase activity = 1.46 and 1.47 (U/mg protein), respectively, whereas the derivative strain PB12AGD^+^ showed an increase of 34% compared to both strains. The cAMP concentration in PB12 increased by 20% compared to the wild-type JM101, whereas the derivative PB12AGD^+^ decreased by 14% and 28% compared to the wild-type JM101 and PB12, respectively.

The previous selection of fast-growing PTS^−^ mutants of *E. coli* in glucose, selected from adaptive laboratory evolution (ALE) experiments, showed an increase in μ in respect to their non-evolved PTS-mutants. Still, the observed μ was always lower than the observed for the wild-type strains. This decrement in growth capabilities was explained by a reduced ATP availability due to the selection of alternative glucose transporters (ABC type) requiring one APT for transport and the further glucose phosphorylation with Glk from ATP, reviewed in [18]. The observed decrement in the [ATP]/[ADP] ratio in a PTS^−^ mutant with a compromised ATP availability for glucose transport, and the lower intracellular level of cAMP due to the absence of EIIA^Glc^ in the derivative PB12AGD^+^, are determinants of its lower growing capabilities compared to the JM101 and the PB12 strains.

### 3.3. Differentially Expressed Genes in Response to the Disruption of the [ATP]/[ADP] Ratio in the Derivative Strain PB12AGD^+^

When *E. coli* transports glucose by the PTS system, the intracellular level of cAMP indicates the presence of glucose in the medium. As glucose is transported by the cell, it results in an inhibition of CyaA. A higher cAMP concentration correlates with a glucose starvation condition [4]. When glucose or other PTS sugars are consumed from the culture medium, the PTS proteins are found in phosphorylated form. P~IIA^Glc^ activates the adenylate cyclase (CyaA), converting ATP to cAMP, forming a complex with the catabolite repressor protein (CRP). The complex CRP–cAMP activates and modulates the expression of several genes repressed by glucose or other PTS sugars [4]. Nevertheless, in a PTS^−^ mutant of *E. coli*, like PB12 (Δ*ptsHIcrr*), the protein EIIA^Glc^ is absent, avoiding the activation of CyaA and the synthesis of cAMP, resulting in the abolition of the catabolite repression mechanism leading the capability to this mutant to consume glucose + acetate and mixtures of glucose-arabinose, glucose-gluconate, and glucose-glycerol simultaneously [11,19].

To assess the impact of the energetic disruption in the mutant PB12AGD^+^ and the lower cAMP concentration with a possible affectation of its role as DNA-binding transcriptional dual regulator coupled to CRP (CRP-cAMP), we analyzed the expression of 96 genes encoding for ATP synthase, glucose and acetate transport, glycolysis and gluconeogenesis, PPP, TCA cycle and glyoxylate shunt, several anaplerotic, respiratory chain, and fermentative pathways genes, sigma factors, and several global regulators (Table 3 and Figure 2 and Figure 3). As previously reported, the wild-type strain JM101 was used as the control to normalize the data using the wild-type RT–qPCR value for that gene. The results showed an increased relative transcription level in all analyzed genes in PB12 and the PB12AGD^+^ mutant compared to JM101 (Table 3). Still, only 55 genes were found upregulated (≥1.7) in PB12 and 34 genes in the mutant PB12AGD^+^.

## 4. Discussion

In the following sections, we discuss the physiological relevance of the differential upregulation on analyzed genes in response to ATP disruption in the PTS^−^ mutants.

### 4.1. Expression of the atpAGD Operon, ATPase Activities, [ATP]/[ADP] Ratio, and Cyclic AMP (cAMP) Concentrations

The total ATPase activity in *E. coli* measures the bacterium’s ability to hydrolyze ATP to ADP and inorganic phosphate. The activity of the F_1_F_0_ ATP synthase is responsible for the synthesis of ATP during oxidative phosphorylation and for hydrolyzing ATP under certain conditions [9]. Uncoupled ATPase activity in the PB12AGD^+^ mutant results from the cloning and expression of the *atpAGD* operon encoding for the F_1_ component of the F_1_F_0_ ATP synthase, increased the overall ATPase activity (U/mgprotein) by ~135% than JM101 and PB12, while these strains showed practically the same activity (Table 2); nevertheless, as expected, the [ATP]/[ADP] ratio in the AGD^+^ mutant was the lower one observed in the analyzed strains. A decreased [ATP]/[ADP] ratio in *E. coli* results from a reduction in the rate of the synthesis of ATP and is considered an indicator of oxidative stress, resulting in a reduction in growth and protein synthesis [20,21].

The relative transcription level of genes encoding the F_1_-ATPase in the plasmid-cloned operon *atpAGD* encoding the α, β, and γ components of the F_1_-ATP synthase showed, as expected, an increased expression by 8–10 X in PB12ADG^+^ than that observed in the strain PB12 (Table 3, Figure 3). The relative transcription level of the *atpAGD* operon is related to the increment of the total ATPase activity detected in the AGD^+^ mutant. As a control of the transcriptional level of the chromosomal *atp* operon, the *atpI* gene encoding for the ATP synthase accessory factor [4] showed a lower expression level than the observed in the PB12 strain (Table 3).

The observed decrement in the [ATP]/[ADP] ratio in the AGD^+^ mutant is associated with the reduction in the intracellular level of cAMP compared to the mutant PB12 (~28%). These results suggest that the decrement in the available ATP resulting from the uncoupled ATPase activity of the F1 component of the F_1_F_0_ ATP synthase has a negative impact on the availability of intracellular cAMP synthesized by adenylate cyclase enzyme CyaA in PB12 and the AGD^+^ mutant. The analysis of the genes with expression regulated by cAMP-CRP, shown in Table 2 (32 genes flagged with *), reports a reduction of 15 genes (~47%), the upregulation of 1 gene, and no change in the expression of 16 genes (50%). These results suggest that despite the reduction in the concentration of cAMP in the *apt* mutant, they do not have a severe effect on the expression of the target genes.

### 4.2. Glucose Transport and Phosphorylation

The previous characterization of the evolved strain PB12 showed that this mutant selected the galactose permease GalP for glucose transport from the periplasm into the cytoplasm and phosphorylated it by Glk from ATP in the Δ*ptsHIcrr* mutant [10,11]. In this study, the relative transcriptional analysis of *ompF, galP, ptsH, ptsG,* and *glk* in strains PB12 and PB12AGD^+^ showed, as expected, no expression of *ptsH* (Hpr), as the *ptsHIcrr* operon was deleted in PB12 by inactivation of the encoding operon in strain JM101 (Table 3). Still, *ptsG* encoding for the inner membrane enzyme IIBC^Glc^ component of PTS did not show significant upregulation in PB12 and PB12AGD^+^ compared to JM101 (Table 3, Figure 2). The expression of *ompF* encoding the porin OmpF was explored as an indicator of glucose diffusion from the extracellular medium into the periplasm. Still, no significant expression was observed in both PTS^−^ mutants. Nevertheless, *galP* was found unregulated in both PTS^−^ mutants, but with a lower relative transcription of ~46% in the PB12AGD^+^ mutant than PB12. As stated, the evolved strain PB12 increased its μ to 0.43 h^−1^ during an ALE experiment conducted to select fast-growing PTS^−^ derivatives, compared to the JM101 PTS^−^ mutant (designated as PB11) with μ = 0.11 h^−1^ [10,11]. The previous fluxomic and transcriptomic analysis in PB12 revealed an increased carbon flux through the first step in the glycolysis by 95%, compared to the JM101 [11], associated with an increased expression of *galP* and *glk* in PB12 (Flores et al. 2005a). Nevertheless, although the strain PB12 showed a higher relative transcription of *galP* and *glk* than the *atp* mutant, the observed q_Glc_ in this mutant increased by 75.3% more than PB12 (Table 2), suggesting a relevant glycolytic activity.

### 4.3. Glycolysis, Gluconeogenic, Pentose Phosphate, and Entner-Doudoroff Pathways

The glycolytic gene *pgi* encoding for glucose-6-P isomerase was upregulated in both PB12 mutants. The upregulation of *pgi* with *galP* and *glk* was proposed to contribute to the increased glycolytic flux observed in the PB12 mutant, which could result in a possibly higher pool of fructose-6-P in both PB12 mutants [11,22]. Upregulation of *pgi* was associated with a reduced carbon flux from glycolysis to PPP and the Entner–Duodoroff pathways [22]. Other glycolytic genes upregulated in PB12 by at least 1.7-fold or higher were *fbaB*, *fbp, gpmA,* and *pykA* (Table 3, Figure 2)*,* encoding, respectively, to fructose-biphosphate aldolase (FbaB), fructose-1-6, biphosphatase 1 (Fbp), 2,3-bisphosphoglycerate-dependent phosphoglycerate mutase (GpmA), and pyruvate kinase 2 (PykA). Although these genes showed a decrease in their relative expression in the *atp* mutant compared to PB12 (Table 3, Figure 2), they support an increased glycolytic flux in this mutant, as suggested by the increased *q_Glc_* value (Table 2).

The PPP genes observed upregulated in both PB12 mutants were *eda, rpiB, talA,* and *tktB,* encoding, respectively, to KHG/KDPG aldolase (Eda), allose-6-phosphate isomerase/ribose-5-phosphate isomerase B (RpiB), and the transketolase 2 (TktB). Nevertheless, *zwf* encoding the NADP+-dependent glucose-6-phosphate dehydrogenase catalyzing the reaction of glucose-6-P + NADP^+^ → 6-P D-glucono-1,5-lactone + NADPH + H^+^ showed an increased relative transcription of 1.96 in PB12, but not in the *apt* mutant (1.25) (Table 3, Figure 2). Despite the observed reduced relative expression of *zwf* in the *apt* mutant but associated with the observed upregulated genes of the PPP, both strains presented an increased carbon flux from glucose-6-P to the PPP.

The expression of the *aptAGD* results in a high turnover of ATP, which has been associated with a strong stimulation of the glycolytic flux resulting in a decrease in the growth rate of the cells [9]. In agreement with this observation, the decrement observed in growth in the *atp* mutant (μ = 0.24 h^−1^) may be associated with lower availability of ATP; the PB12 strains use the GalP to internalize glucose from the periplasm. In the glycolytic pathway, several reactions consume one ATP; phosphorylation of glucose from ATP by Glk (−1 ATP) and conversion of fructose-6-P to fructose-1,6-diP from ATP by PfkA/PfkB (−1 ATP), resulting in the consumption of 2 ATP (-2 ATP). Nevertheless, two reactions involve the synthesis of ATP from ADP: the conversion of 2[3-P-glycerol-P] to 2[3-P-glycerate] by Pgk (+2 ATP), and the conversion of 2[phosphoenolpyruvate] to 2[pyruvate] by PykF/PykA (+2 ATP). Additionally, as the gluconeogenic gene *ppsA* (PpsA) was upregulated in both PB12 strains, it could be considered to be the conversion of at least one pyruvate molecule to phosphoenolpyruvate with 1 ATP. Then, the PTS^−^ mutant PB12 showed the final balance of +1 ATP in the glycolytic pathway from glucose to PEP and the gluconeogenic reaction catalyzed by PpsA. Although the ATP balance in the PB12 AGD^+^ *atp* mutant is the same as that in PB12, the higher ATPase (activity consuming) ATP and lower [ATP]/[ADP] ratio than PB12 indicates the lower growth in the *atp* mutant than in PB12, but, as discussed above, the higher *q_Glc_* showed by the *atp* mutant indicates an increased carbon flux in response to the lower [ATP]/[ADP] ratio.

### 4.4. The TCA Cycle and the Glyoxylate Shunt

The conversion of PEP to oxaloacetate in the reaction catalyzed by phosphoenolpyruvate carboxylase (Ppc), and the conversion of pyruvate to acetyl-CoA (ACoA) catalyzed by the pyruvate dehydrogenase complex (AceE, AceF, and Lpd) feed ACoA into the TCA cycle as substrates of the enzyme citrate synthase (GltA) (Figure 2). At each turn of the TCA cycle, one molecule of ACoA is converted into two molecules of CO_2_, reducing four NAD^+^, NADP^+^, or quinones to NADH, NADPH, and quinol, respectively. The TCA cycle phosphorylates one molecule of GDP to GTP and produces one molecule of fumarate from succinate. Additionally, the TCA cycle includes the glyoxylate shunt, bypassing the reactions of the TCA cycle and eliminating the loss of carbon as CO_2_ [4]. In a biosynthetic direction, carbon from the TCA cycle is withdrawn in two ways: by the decarboxylation of malate to from pyruvate by the malic enzymes MaeA (requiring NAD^+^) and MaeB (requiring NADP^+^), and from oxaloacetate to PEP by the phosphoenolpyruvate carboxykinase (ATP) enzyme (Pck) [4,11].

Transcriptomic analysis of most of the genes encoding enzymes of the TCA cycle and the glyoxylate shunt in the PB12 and PB12AGD^+^ mutants showed an increased relative transcription in these genes than in JM101 (1.7-fold or higher) (Table 3, Figure 2). These results indicate that both strains use the TCA cycle for energy and reducing power. As proposed, the glyoxylate shunt is used under aerobic growing conditions with glucose as the carbon source in the PB12 mutant [11]. Our results indicate that the *apt* mutant maintains this metabolic trait despite the imposed energetic perturbation.

Among the analyzed TCA cycle and the glyoxylate shunt genes in the mutant PB12AGD^+^, *sdhC, fumB,* and *fumC* showed a higher transcription than in PB12 (Table 3, Figure 2). *sdhC* encodes a succinate:quinone oxidoreductase membrane protein, SdhC, which is a component of the four-subunit succinate dehydrogenase (SQR) enzyme [4]. SdhC is the large subunit of cytochrome b556, and the quinone binding (Qp) site resides in the interface between SdhB, SdhC, and SdhD [4,23,24]. The maximal expression of SQR was reported during aerobic growth, and the oxidation of succinate to fumarate concomitant was catalyzed with the reduction of ubiquinone to ubiquinol. SQR is essential in cellular metabolism, connecting the TCA cycle with the respiratory electron transport chain. As part of the TCA cycle, succinate is oxidized to fumarate by SQR, and electrons are transferred to the membrane quinone pool for entry into the electron transport chain, but do not contribute to the PMF [4,25].

FumA (fumarase A) is one of the three fumarase isoenzymes in *E. coli* (encoded by *fumA, fumB*, and *fumC*), participating in the TCA cycle catalyzing the reversible conversion of (*S*)-malate to fumarate + H_2_O [4]. FumA was reported as the most abundant fumarase under aerobic growth conditions, and its expression is controlled by the transcriptional aerobic/anaerobic regulators ArcA, whereas Fnr represses *fumA* [26]. As our results showed that the expression of *fumA* in the *atp* mutant practically duplicated the observed in PB12, this difference could be considered an indicator of an active role in the TCA cycle in this mutant.

The sensor histidine kinase ArcB represses genes encoding enzymes that catalyze the reaction of the TCA cycle. The DNA-binding transcriptional dual regulator ArcA is part of the ArcAB two-component system involved in response to respiratory conditions, and growth changes with a role in the anaerobic repression of genes associated with aerobic metabolism [4]. Previous results on the mutant PB12 showed that this strain possesses an ArcAB^−^ phenotype due to a point mutation resulting in the mutation Tyr_71_Cys of ArcB [11]. This mutation was proposed to participate in a disulfide bridge formation between two subunits of ArcB, resulting in an ArcAB with an inactivated or diminished function in this strain that has been proposed to result in the upregulation of the TCA cycle and the glyoxylate shunt genes during aerobic growth despite the upregulation of these regulator proteins in PB12 [11]. As observed in Table 3, *arcB* was found upregulated in both PB12 mutants, suggesting the proposed role of this regulator also in the *atp* mutant.

### 4.5. Fermentation, Acetate Production, and Utilization Genes

The fermentative gene *ldhA* (involved in lactate production from pyruvate), the acetate production from pyruvate, and acetate utilization genes *pflD, poxB,* and *acs* were found upregulated in PB12, as in the PB12AGD^+^ mutants, than in JM101 (Table 3, Figure 2), suggesting that both mutants produce fermentation products. PB12 has been reported to produce smaller amounts of lactate and acetate in the late log phase [11]. Our results led us to propose that the *atp* mutant maintains this metabolic trait. Acetate formation from pyruvate involves the conversion of pyruvate to ACoA+ formate with PflB and the conversion of pyruvate to ACoA with AceE (pyruvate dehydrogenase). Then, ACoA is converted to acetyl-P by Pta, and the conversion of acetyl-P + ADP to acetate + ATP is performed with AckA (Figure 3). The second pathway involves the conversion of pyruvate to acetate by PoxB in a reaction coupled to the electron transport chain via ubiquinone [4]. The acetate metabolism by PoxB has been proposed to be less efficient than the pathway with AceE.

Nevertheless, the PoxB pathway is relevant in synthesizing acetate under aerobic conditions [4,27], and was identified as the main pathway for acetate production in the stationary phase [4,28]. Previously, it was proposed for the PB12 strain that acetate produced by PoxB could be a possible autoinducer of the *acs* operon and the glyoxylate shunt genes that were upregulated in both PTS^−^ mutants analyzed [11], resulting in a strategy proposed previously to the strain PB12 to convert acetate to AcCoA with Acs and PoxB, a characteristic that maintains the PB12AGD^+^ mutant. Remarkably, the previous inactivation of *poxB* in PB12 did not affect growth under aerobic culture conditions, supporting the proposition that PoxB is this mutant’s primary source of ACoA [11]. ACoA synthetase (AMP-forming) encoded with *acs* is one of the two ways by which activates acetate to acetyl-CoA in a pathway working in an anabolic way, scavenging acetate in the medium.

The expression of *poxB* and *acs* is an RpoS (upregulated in both PTS^−^ mutants, Table 3) dependent promoter [11,29], and its activity is induced by growing on acetate [4,30,31,32].

### 4.6. The Respiratory Chain

The respiratory chain of *E. coli* is comprised of NADH dehydrogenase I (NADH-I) and fumarate reductase (FDR). NADH-I, encoded by the *nuo* operon (*nuoABCDEFGHIJKL*), represents the primary dehydrogenase in the respiratory chain in *E. coli*, and appears to be the only H^+^-pumping dehydrogenase in the respiratory chain. This multienzyme complex catalyzes the transfer of two electrons from NADH to quinone, coupled with the translocation of four protons across the membrane, contributing to the proton-motive force (PMF) required for the synthesis of ATP [4,33,34]. Transcriptomic analysis of the genes *nouABCEF* and *nuoMN* in PB12 showed a slightly increased transcriptomic level in this mutant and no changes in PB12 AGD^+^ (Table 3, Figure 3). These genes comprise the soluble fragment (NouEFG) involved in the oxidation of NADH, and constitute the electron input component of the complex, the amphipathic connecting section including the NuoBCD and NuoI, and the membrane fragment composed of NouA, H, J, K, L, and NuoMN [4,33].

UbiE (encoded by *ubiE*) is a C-methyltransferase that catalyzes reactions in both ubiquinone (Q) and menaquinone (vitamin K_2_ [MK]) biosynthesis. The isoprenoid quinone ubiquinone (coenzyme Q) is an essential component in the respiratory electron transport chain, whereas MK is an isoprenoid naphthoquinone functioning as a redox mediator in *E. coli* electron transfer chains [35]. The transcriptomic analysis of this gene showed an upregulation of this gene in PB12 and a slight upregulation in the apt mutant (Table 3).

The gene *napA* encodes for the molybdoprotein subunit of the periplasmic nitrate reductase Nap, and is involved in energy conservation during anaerobic bacterial growth [36]. This protein was shown to be upregulated in the two PB12 mutants. Still, this result is intriguing because the expression of *napA* is induced in anaerobiosis with Fnr, and lower nitrate concentrations are mediated by the transcriptional regulator NarP. Nap was reported to support anaerobic respiration in various carbon sources at the expense of lower nitrate concentrations [37]. The gen *narG* was found also upregulated in both PB12 mutants. This gene encodes the α subunit of the nitrate reductase A (NRA), involved in the anaerobic respiratory chain with the NADH dehydrogenase transferring electrons from NADH to nitrate coupled to the generation of a PMF across the cytoplasmic membrane in *E. coli* [4]. Expression of NRA was observed in response to high levels of nitrate in the environment [38].

Upregulation of aerobic- and anaerobic-respiratory genes in both PB12 mutants indicates increased respiratory capabilities compared to the parental strain JM101, coupled with a possible higher availability of electron donor compounds from the TCA cycle. Additionally, the upregulation of several analyzed genes results from the proposed ArcAB^−^ phenotype and the upregulation of the transcriptional dual regulator Fnr. ArcA is a repressor of the *cyoABCDE*, the *nuoABCDEFGHIJKL* operons, and *nadH* [4]. Upregulation of these respiratory genes in both PB12 mutants was also proposed due to the ArcAB^−^ phenotype as proposed for the upregulation of the TCA cycle and the glyoxylate shunt genes. Additionally, Fnr induces the expression of the *fdrABCD* operon, *napA*, and *narGI* [4], and this transcriptional regulator was upregulated in these mutants (Table 3, Figure 3). The observed upregulation of the aerobic-respiratory genes is associated with an increased value of q_O2_ in the mutant PB12AGD^+^ (1.19 mmol/g_dcw_ h), an increase of 240% compared to the PB12, indicating that this derivative has increased respiratory activity in response to the modification of the [ATP]/[ADP] ratio.

### 4.7. Regulatory Proteins and Sigma Factors

We determined the relative transcription of several genes encoding regulatory proteins. As Table 3 shows, the transcription of these genes in both PB12 mutants increased compared to the parental JM101.

The Cra (FruR) regulator (DNA-binding transcriptional dual regulator) modulates the direction of the carbon flow through different metabolic pathways [4]. According to the EcoCyc database, Cra is proposed to regulate 61 transcriptional units, including the negative regulation of glycolytic and gluconeogenic enzymes (genes included in this studio *eno, fbaB, gapA, glk, pgk, pykF, tpiA, pckA, ppsA*), PPP (*zwf, eda*), the TCA cycle, and glyoxylate shunt enzymes (*aceEF, acnB, lpd*) (*icdA,* positively regulated), acetate production and utilization (*poxB*, positively regulated), respiratory genes (*cyoBCDE*), and genes encoding the Entner–Doudoroff pathway [4].

The GlcC regulator (GlcC-glycolate DNA-binding transcriptional dual regulator, encoded with *glcC*) is proposed to control the expression of genes involved in utilizing glycolate as the sole carbon source [39]. The transcriptional regulator Gycolate-GlcC positively controls the transcription of the operon *glcDEFGBA*, resulting in the expression of the glycolate dehydrogenase (GlcEFD) and the malate synthase G (GlcB) involved in the glyoxylate shunt [4].

The gen *ihfA* encodes the integration host factor subunit α (IHFα or IFHA). *E. coli* IHF is a complex conformed by the subunits IHFα and IHFβ. IHFα is the subunit responsible for DNA binding and is essential for the induction of resistance to extreme acidic pH in *E. coli* [40]. IHF was also involved in controlling the expression of several regulators and genes involved in cellular processes in *E. coli*. The EcoCyc database reports 106 transcriptional units controlled by IHF including several genes analyzed in this study like *ompC* (controlled negatively), the TCA cycle and the glyoxylate shunt (*sucABCD,* controlled negatively; *aceBAK* and *glcB* controlled positively), fermentative and acetate utilization (*pflB* and *acs,* controlled negatively), respiratory genes (*nuoABCFMN*, *ndh, narG*, all controlled negatively), and Fnr (controlled positively) [4].

The DNA-binding transcriptional repressor IclR (coded by *iclR*) is involved in adaptive responses in bacteria. In *E. coli,* this repressor is reported to control the expression of genes involved in the glyoxylate shunt positively [41], such as *aceBAK* (included in this study). FadR, the DNA-binding transcriptional dual regulator encoded by *fadR,* is the fatty acid degradation regulator that negatively controls fatty acid and lipid metabolisms [42] and acetate metabolism [43]. Acetate transport, oxidation, and incorporation into macromolecules were observed to increase (5X) in a *fadR*^−^ mutant during growth on succinate as a carbon source, associated with an increased level of glyoxylate shunt in this mutant [43,44]. As FrdR activates the expression of *iclR*, it has been proposed as an indirect upregulation of *aceBAK* genes involved in the glyoxylate shunt [11,44].

Finally, the DNA-binding transcriptional dual regulator Fnr (encoded by *fnr*) is a primary transcription regulator controlling the transition from aerobic to anaerobic growth through the control of more than 140 genes [4], including acid resistance, chemotaxis, cell structure, and several biosynthetic processes [45]. There are 142 transcriptional units controlled with Fnr, including the negatively controlled TCA cycle genes *aceEFB, sdhCDAC, sucABCD*, and *lpd* genes, and the respiratory genes *cyoBCDE* and *narG*. Additionally, Fnr positively controls the expression of ArcA [4]. As the cellular concentration of Fnr is the same under both aerobic and aerobic growth conditions, its activity is regulated by oxygen concentration [46]. In addition to the direct role as a transcriptional factor, Fnr can act indirectly through the increased expression of ArcA and the target genes [45].

Fnr acts as a dimmer; in the presence of oxygen, the dimeric structure is converted to a monomeric form, reducing its ability to bind target DNA. Additionally, depending on the growth conditions, the proposed number of targets for Fnr and other global regulators varies (e.g., growing on a minimal M9 medium supplemented with glucose vs. growing in LB medium) [45].

The expression of genes encoding the sigma factors *rpoD, rpoE, rpoH*, *rpoN,* and *rpoS* showed an increased expression in both PB12 mutants compared to the wild-type strain. The genes *rpoA, rpoC*, and *rpoZ* encoding the RNA polymerase subunits α, β’, and ω, respectively, were found unchanged in both strains (*rpoA*), unregulated only in PB12 (*rpoC*) and upregulated in both strains (*rpoZ*) (Table 3). The increased transcription of *rpoS* and *rpoD* has been proposed to indicate the exposition to nutritional stress in wild-type *E. coli* [47,48]. In strains lacking the *crr* gene, like the PB12 mutants, the absence of the EII^Glc^ enzyme led to higher levels of RpoS even during exponential growth, allowing the transcription of several central carbon metabolism genes. This condition led to the proposal of σ^38^ as a second vegetative sigma factor [4]. *rpoD* was the unique sigma factor more highly expressed in the AGD^+^ than in PB12, showing an increment of 75% in the relative transcription level. The increased level of σ^70^ in the AGD^+^ mutant and the reduced expression level of σ^38^ compared to the PB12 (a decrement of ~66%) suggest the primary role σ^70^ under a nutrient limitation condition as a strategy of the cell to promote the upregulation of growing genes in this mutant under this nutritional scavenging condition, resulting in the reduced [ATP]/[ADP] ratio, a lower cAMP concentration, and a lower μ value compared to the PB12.

The DNA binding transcriptional dual regulator *narL* was slightly upregulated in both PB12 mutants. NarL is a transcriptional regulator of many aerobic, electron transport, and fermentation-related genes in response to higher nitrate and nitrite levels. According to the RegulonDB [49], this regulator negatively controls the expression of the studied genes *frdB, napA* (respiratory chain), and fumB (TCA cycle), but positively controls the expression of respiratory genes *narG, nuoABCFMN,* and *poxB* (acetate metabolism and utilization).

Remarkably, the sigma D regulator *rsd,* the DNA-binding transcriptional dual regulators *soxS* and *soxS,* and the bifunctional (p)ppGpp synthase/hydrolase, *spoT,* showed the highest relative transcription in the AGD^+^ mutant than in PB12. The gene *rsd* increased in ~115X, *soxS* above 3000X, *soxR* above 380X, and *spoT* by ~300X (Table 3). Rsd is an alternative σ factor-dependent transcription. Rsd was proposed to be present and regulates gene expression from the early exponential to stationary phase [50], but levels increase when exponential growth decreases, increasing the expression of RpoS-dependent promoters during the stationary phase [51]. The dephosphorylated form of HPr protein sequesters Rsd, leading RpoD to bind to the core of the RNA polymerase, transcribing housekeeping genes during exponential growth [4]. As Hpr is absent in the mutants PB12 (Δ*ptsHIcrr*) and the higher relative transcription of the gene *rsd*, Rsd should decrease the expression of σ^70^-dependent promoters. Still, this sigma was upregulated during exponential growth in the AGD^+^ mutant. Nevertheless, [50] reported that the 6S RNA, which also increases its concentration during growth, minimizes the effect of Rsd on RNA polymerase σ^70^, leading to the expression of σ^70^-dependent promoters, as observed in the AGD^+^ mutant.

### 4.8. Expression of Stress Response Genes

Disruption of normal [ATP]/[ADP] ratio in *E. coli* by uncoupling the cytoplasmic activity of the F_1_ subunit affects the normal balance of ATP an NADH, significantly affecting the redod balance of the cell, resulting in the accumulation of reactive oxygen species and genes involved in respiration [52]. The transcriptional factors SoxR and SoxS control a superoxide response regulon. SoxR induces *soxS* expression, and SoxS activates the transcription of at least 14 genes of the regulon, including genes involved in the elimination of oxidative stress, repair or compensate for damage, or the induction of metabolic functions that led the cell to contend with oxidative stress [53,54].

The disruption of the [ATP]/[ADP] ratio in the AGD^+^ mutant triggered a strong response to oxidative stress. Among the analyzed genes in both PB12 mutants, *zwf* and *fumC* are reported to be induced by *soxS* and *soxR* in response to oxidative stress conditions. Induction of *zwf* results in the regeneration of NADPH, and induction of *fumC* replaces the O_2_^• −^ sensitive fumarase [4,54]; nevertheless, the expression of these genes was lower in the AGD^+^ mutant than in PB12 (Table 3). The higher level of relative expression detected for *soxR* and *soxS* in the AGD^+^ mutant suggests that this strain is under some stress condition probably associated with the higher oxygen consumption rate observed = 1.17 mmol/g_DCW_.h, a higher value than the observed for PB12 (an increase of 138X), as it was reported that reactive oxygen species are produced as a byproduct of aerobic respiration [55,56].

Finally, *spoT* encoding the bifunctional (p)ppGpp synthase/hydrolase SpoT was the highest upregulated gene in the AGD^+^ mutant (~257X), compared to the PB12. SpoT is a crucial enzyme involved in the stringent response in *E. coli* [4], which is a global regulatory system activated under conditions of nutrient or energy starvation or other environmental stress, resulting in the modification of gene expression, metabolism, and cellular processes under these conditions [4,57]. In *E. coli*, the stringent response is mediated by the concentration of guanosine 3′,5′-bis(diphosphate) and guanosine 3′-diphosphate,5′-triphosphate, known as (p)ppGpp that modulates the affinity of sigma factors for the RNAp core. Environmental stress results in variable concentrations of (p)ppGpp concentrations; higher levels of (p)ppGpp result in higher expression of rpoS by the direct stimulation of the transcription of other genes related to stress protection [4,57].

The higher transcription level of *spoT* in the AGD^+^ mutant could be considered (as stated above) an indicator that this mutant is under energy starvation. In this condition, the synthetase activity of SpoT (together with RelA) synthesizes ppGpp [58], increasing its concentration and regulating several cellular processes to adapt and live under stress. This physiological condition suggests a relevant role of RpoS. Nevertheless, as shown in Table 3, the expression of this sigma factor showed a lesser value than PB12. Remarkably, the analysis of the correlation of the (p)ppGpp levels in diverse *E. coli* K12 strains suggests that there is not necessarily a strong correlation with concentration with the concentration of σ^38^ [59].

### 4.9. Physiological Significance of [ATP]/[ADP] Ratio Disruption in E. coli PTS^−^

Overexpression of genes encoding the ATP-hydrolyzing component of the ATPase encoded with the *atpAGD* operon in several strains of *E. coli* glucose uptake increase the carbon flux through the glycolytic pathway and compensate for the loss of ATP [9,60]. ATP wasting has been used as a suitable strategy for overproducing valuable compounds in *E. coli* and other microorganisms like *Saccharomyces cerevisiae* and succinic acid, recombinant protein, lactate, acetoin, ethanol, and 2,3-butanediol [8,61,62].

The PTS mutant of *E. coli* PB12 has been used as a suitable host strain for the overproduction of shikimic acid (SA), a precursor for synthesizing valuable chemicals [63]. The results presented in this contribution suggest an increased glycolytic carbon flux and a robust response to several stress conditions to contend against the disruption of the [ATP]/[ADP] ratio in the PTS^−^ *atp* mutant. These traits make this mutant a genetic background of interest to evaluate its capacity to produce metabolites such as SA.

## 5. Conclusions

The disruption of the balance of energy production and utilization in the PTS^−^ strain of *E. coli* PB12 by plasmid expression of the *atpAGD* operon encoding the soluble portion of ATP synthase F_1-_ATPAse (PB12 AGD^+^ mutant) resulted in a reduction of the [ATP]/[ADP] ratio and the intracellular level of cAMP. The mutant grows slowly in glucose, but showed a higher *q_Glc_* and lower biomass yield than the strain PB12. Higher relative transcription of *galP* and *pgi* genes is associated with these traits, suggesting an increased carbon flux through glycolysis and the Entner–Duodoroff pathway. The upregulation of the acetate transport and genes involved in acetate synthesis and utilization, with a lower acetate concentration detected in supernatant cultures, supports the coutilization of acetate and glucose to increase the proposed carbon flux in this strain in response to the reduction in the [ATP]/[ADP] ratio.

Upregulation of most genes encoding the TCA cycle and the glyoxylate shunt indicates that the AGD^+^ mutant employs these pathways for energy and the reduction in power supply growth in glucose. The upregulation of several respiratory genes indicates increased respiratory capabilities, coupled possibly with the increased availability of electron donor compounds from the TCA cycle, resulting in an increased respiratory capability of 240% in the *apt* mutant than in the PB12. The upregulation of genes involved in stress response in the *apt* mutant, including *soxR* and *soxS* (superoxide response regulon), probably in response to a higher oxygen consumption observed in this mutant, makes sense with the increased respiratory capability. The upregulation of *spoT* (~357X in the *apt* mutant) confirms the severe stress condition resultant in response to the severe energetic disruption imposed in this mutant.

Finally, the reduction in the intracellular concentration of cAMP in the *atp* mutant resulted in a reduced number of upregulated genes than in PB12, showing a downregulation of the expression of 15 genes regulated by cAMP-CRP compared to the PB12 strain, suggesting that the mutant has a very robust genetic background despite the severe disruption in its energetic level. Our results provide valuable information to understand the capacity of this evolved mutant PB12 to adapt to a drastic energetic perturbation.

## Figures and Tables

**Figure 1 biotech-13-00010-f001:**
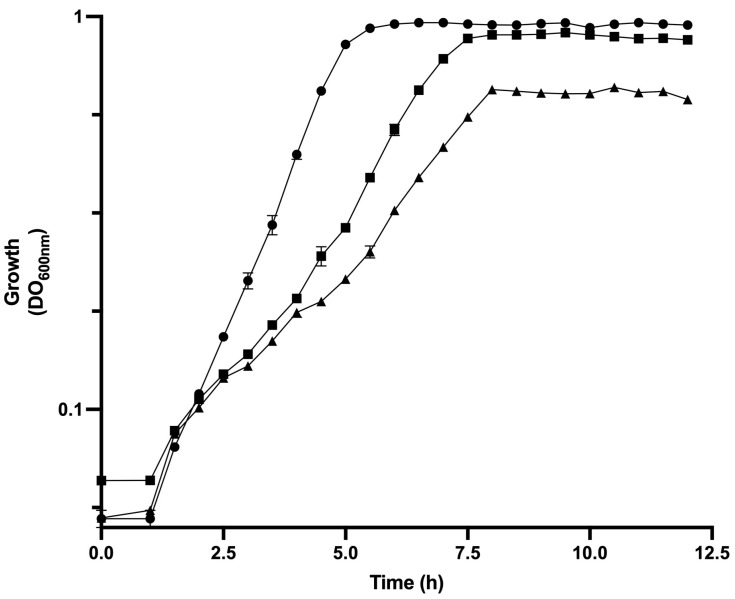
Growth curves of the parental *E. coli* JM101 strain (●) and the PB12 (■) and PB12AGD^+^ (▲) derivatives conducted in bioreactors. These aerobic cultures were performed in M9 supplemented with glucose (2 g/L). All experiments were performed in duplicate, and the reported values represent the mean values of at least four independent experiments. The difference in measured metabolite concentrations among independent experiments was in the range of 1–5%.

**Figure 2 biotech-13-00010-f002:**
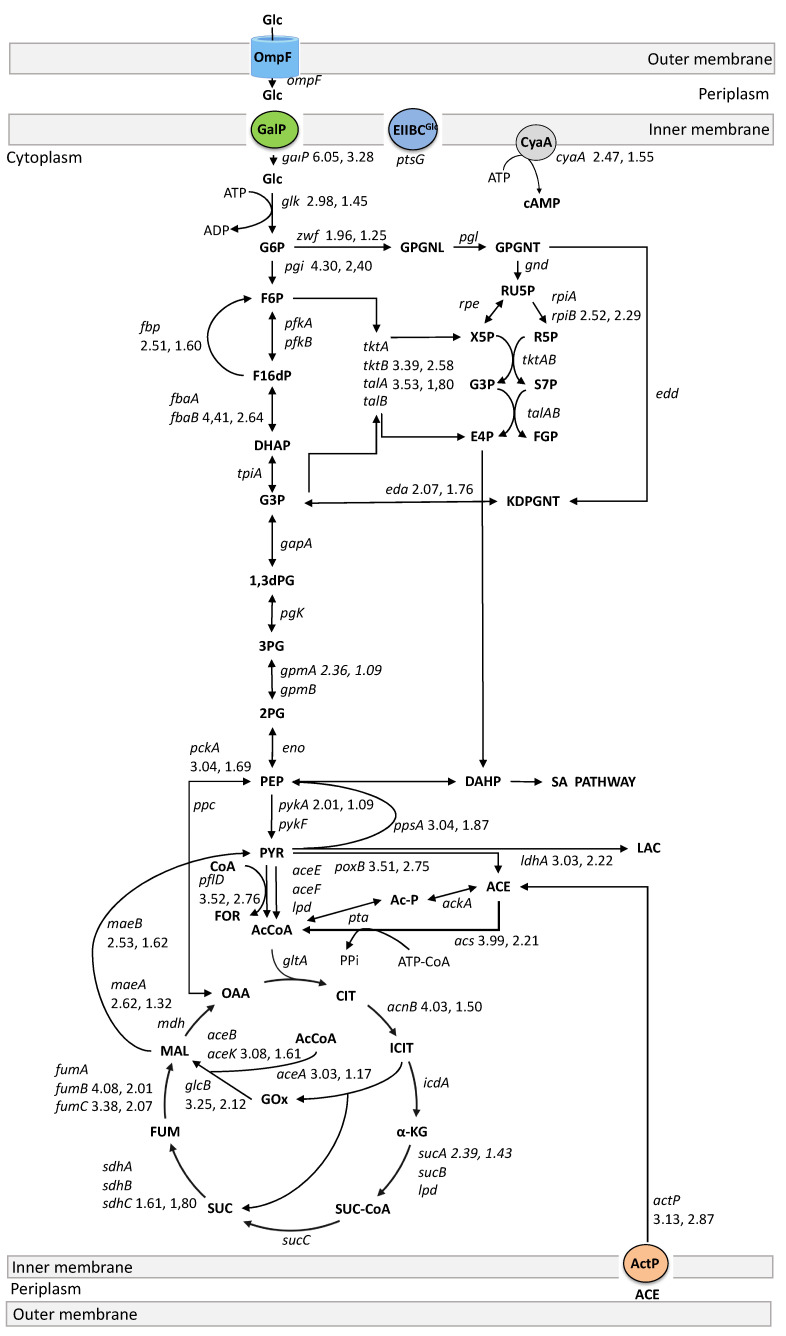
Relative transcript levels of genes involved in glucose transport, carbon central metabolism, and fermentation pathways. RT–PCR values of those upregulated genes (1.7-fold or higher) are shown beside the gene’s name in parenthesis: The first value for PB12 and the second for PB12AGD^+^. The relative gene transcription value for JM101 is always equal to 1. Metabolite abbreviations: Glc, glucose; G6P, glucose-6-phosphate; F6P, fructose-6-phosphate; P1,6dP, fructose-1,6-biphosphate; DHAP, dihydroxyacetone phosphate; G3P, glyceraldehyde 3-phosphate; 1,3-dGP, 1,3-biphosphoglycerate; 3PG, 3-phosphoglycerate; 2PG, 2-phophoglycerate; PEP, phosphoenolpyruvate; PYR, pyruvate; 6PGLN, 6-phosphoglucono-δ-lactone; 6PGNT, 6-phophogluconate; KDPGNT, 2-keto-3-deoxy-6-phosphogluconate; RU5P, ribulose-5-phosphate; R5P, ribose-5-phosphate; X5P, xylulose-5-phosphate; S7P, sedoheptulose-7-phosphate; E4P, erythrose-4-phosphate; F6P, fructose-6-phosphate; AcCoA, acetyl coenzyme A; CoA, coenzyme A; Ac-P, acetyl phosphate; ACE, acetate; LAC, lactate; FOR, formate; CIT, citrate; ICIT, isocitrate; GOx, glyoxylate; α-KG, α -ketoglutarate; SUC-CoA, succinyl-coenzyme A, SUC, succinate; FUM, fumarate; MAL, malate; OAA, oxaloacetate; SA, shikimic acid; DAHP, 3-deoxy-D-arabino-heptulosonate 7-phosphate; cAMP, cyclicAMP. The corresponding names of the genes and the complete RT–qPCR values for all analyzed genes are shown in Table 3.

**Figure 3 biotech-13-00010-f003:**
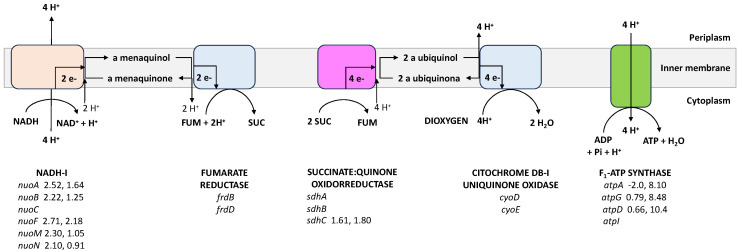
Relative transcript levels of genes involved in the respiratory chain and oxidative phosphorylation in PB12 and PB12AGD^+^ derivatives. RT-PCR values of those upregulated genes (1.7-fold or higher) are shown beside the gene’s name in parenthesis: The first value for PB12 and the second for PB12AGD^+^. The relative gene transcription value for JM101 is always equal to 1. NADH-I, NADH:quinone oxidoreductase I; FUM, fumarate, SUC, succinate. The corresponding names of the genes and the complete RT-qPCR values for all analyzed genes are shown in Table 3. The reaction mechanisms were adapted from the EcoCyc database (https://ecocyc.org, accessed on 8 January 2024) [4].

**Table 1 biotech-13-00010-t001:** Plasmids and bacterial strains used in this study.

Plasmids	Relevant Features	References
pTrc99A	Cloning vector under the *trc* promoter. Carries *bla* and *lacI^q^* genes. Replication origin from pBR322.	[12]
pTrc*atpAGD*	*atpAGD* operon cloned under the control of *trc* promoter into the pTrc99A vector.	[6]
**Bacterial strains**		
JM101	F′ traD36 *proA^+^ proB^+^ lacI^q^ lacZ*Δ*15/supE thi* Δ*(lac-proAB).*	[14,15]
PB12	JM101 Δ*ptsHptsIcrr*::*Km*, PTS^−^ Glc^+^.	[10,11]
PB12AGD^+^	PB12 transformed with pTrca*tpAGD.*	This study
PB12pTrc99A	PB12 transformed with pTrc99A.	This study

**Table 2 biotech-13-00010-t002:** Physiological parameters determined from bioreactor cultures of JM101, PB12, and PB12AGD^+^ strains.

Strain	µ (h^−1^)	*q*_Glc_ (g_Glc_/g_DCW_ h)	Y_X/Glc_ (g/g)	*q*_Ace_ (g/g_DCW_ h)	*q_B_*_ase_ (mmol/g_dcw_ h)	*q_O2_* (mmol/g_Dcw_ h)	[ATP]/[ADP] Ratio	Total ATPase Activity (U/mg_protein_)	Intracellular cAMP (pmol/mg Protein)
JM101	0.71 ± 0.04	1.74 ± 0.04	0.41	0.10 ± 0.01	8.41 ± 0.36	1.58 ± 0.06	7.32 ± 0.14	1.46 ± 0.05	24.94 ± 1.25
PB12	0.43 ± 0.02	0.81 ± 0.08	0.53	0.02 ± 0.00	5.60 ± 0.02	0.5 ± 0.01	2.53 ± 0.03	1.47 ± 0.01	29.92 ± 1.98
PB12AGD^+^	0.24 ± 0.04	1.42 ± 0.07	0.17	0.06 ± 0.00	7.34 ± 0.07	1.19 ± 0.03	1.20 ± 0.01	1.97 ± 0.01	21.44 ± 2.02

All data are shown in the average obtained from four independent experiments and SD values. The difference in measured metabolite concentrations among independent experiments was in the range of 1–10%. μ = specific growth rate; *q*_Glc_ = specific glucose consumption; Y_X/Glc_ = rate yield biomass/substrate; *q*_Ace_ = specific production rate of acetic acid; *q*_base_ = base addition rate; *q*_O2 =_ oxygen consumption specific rate; g_DCW_ = grams of dry cell weight. Data analysis and calculations were performed as described previously [16]. Linearizations were made to obtain apparent biomass on the substrate (Y_X/S_) yield. Correlation values for linearizations in all experiments were >0.95, allowing comparisons between them. The yield value was used to calculate the specific productivity and consumption rates on the exponential phase.

**Table 3 biotech-13-00010-t003:** Relative transcription levels determined using RT-qPCR for several groups of genes in strains PB12 and PB12AGD^+^.

Pathway, Group of Genes or Cellular Process	Encoded Protein ^1^	Relative Transcription Levels as 2^−^ ^ΔΔCt^	
		PB12	PB12AGD^+^
ATP synthase			
*atpI*	Subunit I	1.61 ± 0.18	0.69 ± 0.05
*atpA*	Subunit α	−2.0 ± 0.03	8.10 ± 2.82
*atpG*	Subunit γ	0.79 ± 0.17	8.48 ± 1.60
*atpD*	Subunit β	0.66 ± 0.11	10.4 ± 1.89
Glucose transport			
*galP **	Galactose:H^+^ symporter	6.05 ± 0.63	3.28 ± 0.33
*ompF **	Outer membrane porin F	0.54 ± 0.08	0.44 ± 0.09
*ptsH **	Phosphocarrier protein Hpr	0	0
*ptsG **	Glucose-specific PTS enzyme IIBC component	1.53 ± 0.22	0.89 ± 0.23
Acetate transport			
*actP (yjcG) **	Acetate/glycolate:cation symporter	3.13 ± 0.07	2.87 ± 0.35
Glycolysis and gluconeogenesis			
*Eno*	Enolase	0.66 ± 0.07	0.33 ± 0.03
*fbaA **	Fructose-bisphosphate aldolase class II	0.64 ± 0.13	0.21 ± 0.09
*fbaB*	Fructose-bisphosphate aldolase class I	4.41 ± 0.94	2.64 ± 0.59
*fbp*	Fructose-1,6-bisphosphatase 1	2.51 ± 0.00	1.60 ± 0.02
*gapA **	Glyceraldehyde-3-phosphate dehydrogenase	1.04 ± 0.16	0.47 ± 0.04
*glk*	Glucokinase	2.98 ± 0.31	1.45 ± 0.30
*gpmA*	2,3-bisphosphoglycerate-dependent phosphoglycerate mutase	2.36 ± 0.67	1.09 ± 0.06
*gpmB*	Putative phosphatase	1.59 ± 0.05	1.42 ± 0.05
*pfkA*	6-phosphofructokinase 1	0.96 ± 0.03	0.48 ± 0.04
*pgi*	Glucose-6-phosphate isomerase	4.30 ± 0.35	2.40 ± 0.25
*pgk **	Phosphoglycerate kinase	0.77 ± 0.04	0.55 ± 0.02
*ppc*	Phosphoenolpyruvate carboxylase	1.53 ± 0.02	1.63 ± 0.11
*pykA*	Pyruvate kinase 2	2.01 ± 0.24	1.09 ± 0.06
*tpiA*	Triose-phosphate isomerase	1.45 ± 0.01	0.68 ± 0.02
Pentose phosphate pathway			
*eda*	KHG/KDPG aldolase	2.07 ± 0.18	1.76 ± 0.09
*gnd*	6-Phosphogluconate dehydrogenase, decarboxylating	1.29 ± 0.49	0.63 ± 0.10
*rpiA*	Ribose-5-phosphate isomerase A	1.63 ± 0.40	1.39 ± 0.06
*rpiB*	Ribose-5-phosphate isomerase B	2.52 ± 0.49	2.29 ± 0.55
*talA*	Transaldolase A	3.53 ± 0.37	1.80 ± 0.32
*talB*	Transaldolase B	1.49 ± 0.05	0.76 ± 0.25
*tktB*	Transketolase 2	3.39 ± 0.06	2.58 ± 0.57
*zwf*	NADP^+^-dependent glucose-6-phosphate dehydrogenase	1.96 ± 0.24	1.25 ± 0.08
TCA cycle and the glyoxylate shunt			
*aceB **	Malate synthase A	1.56 ± 0.00	0.78 ± 0.11
*aceA **	Isocitrate lyase	3.03 ± 0.32	1.17 ± 0.03
*aceK **	Isocitrate dehydrogenase kinase	3.08 ± 0.20	1.61 ± 0.04
*aceE **	Pyruvate dehydrogenase	1.36 ± 0.25	0.78 ± 0.11
*aceF **	Pyruvate dehydrogenase E2 subunit	1.32 ± 0.25	0.93 ± 0.15
*acnB **	Aconitate hydratase B	4.03 ± 0.59	1.50 ± 0.12
*fumA **	Fumarase A	0.77 ± 0.03	1.44 ± 0.03
*fumC **	Fumarase C	3.38 ± 1.01	2.07 ± 0.69
*fumB **	Fumarase B	4.08 ± 0.70	2.01 ± 0.31
*glcB*	Malate synthase G	3.25 ± 0.18	2.12 ± 0.22
*icdA*	Isocitrate dehydrogenase	1.16 ± 0.31	0.39 ± 0.01
*lpd **	Lipoamide dehydrogenase	1.56 ± 0.16	1.19 ± 0.08
*mdh **	Malate dehydrogenase	1.48 ± 0.17	0.75 ± 0.15
*sdhC **	Succinate:quinone oxidoreductase, SdhC	1.61 ± 0.15	1.80 ± 0.20
*sdhA **	Succinate:quinone oxidoreductase, FAD binding protein	0.93 ± 0.01	1.37 ± 0.19
*sdhB **	Succinate:quinone oxidoreductase, iron-sulfur cluster binding protein	1.37 ± 0.27	0.91 ± 0.06
*sucA **	2-Oxoglutarate decarboxylase, thiamine-requiring	2.39 ± 0.44	1.43 ± 0.48
*sucB **	2-Oxoglutarate dehydrogenase E2 subunit	1.17 ± 0.26	0.84 ± 0.31
*sucC **	Succinyl-CoA synthetase subunit β	1.41 ± 0.01	1.14 ± 0.25
Anaplerotic genes			
*maeA* (*sfcA*)	Malate dehydrogenase (oxaloacetate-decarboxylating)	2.62 ± 0.00	1.32 ± 0.02
*maeB*	Malate dehydrogenase (oxaloacetate-decarboxylating) (NADP^+^)	2.53 ± 0.47	1.62 ± 0.43
*pckA **	Phosphoenolpyruvate carboxykinase (ATP)	3.04 ± 0.69	1.69 ± 0.07
*ppsA*	Phosphoenolpyruvate synthetase	3.04 ± 0.29	1.87 ± 0.08
Respiratory chain			
*cyoD **	Cytochrome bo_3_, subunit 4	1.02 ± 0.03	0.63 ± 0.03
*cyoE*	Heme O synthase	1.41 ± 0.31	0.68 ± 0.00
*frdB*	Fumarate reductase iron-sulfur protein	1.49 ± 0.09	0.79 ± 0.08
*frdD*	Fumarate reductase membrane protein FrdD	1.63 ± 0.24	1.16 ± 0.20
*napA*	Periplasmic nitrate reductase subunit NapA	3.83 ± 0.08	2.76 ± 0.12
*narG*	Nitrate reductase A subunit α	4.22 ± 0.58	3.51 ± 0.94
*ndh*	NADH:quinone oxidoreductase II	1.74 ± 0.03	1.10 ± 0.21
*nuoA*	NADH:quinone oxidoreductase subunit A	2.52 ± 0.09	1.64 ± 0.18
*nuoB*	NADH:quinone oxidoreductase subunit B	2.22 ± 0.72	1.25 ± 0.65
*nuoC*	NADH:quinone oxidoreductase subunit CB	1.60 ± 0.23	1.50 ± 0.20
*nuoF*	NADH:quinone oxidoreductase subunit F	2.71 ± 0.24	2.18 ± 0.35
*nuoM*	NADH:quinone oxidoreductase subunit M	2.30 ± 0.32	1.05 ± 0.16
*nuoN*	NADH:quinone oxidoreductase subunit N	2.10 ± 0.65	0.91 ± 0.01
*ubiE*	Bifunctional 2-octaprenyl-6-methoxy-1,4-benzoquinol methylase and demethyl menaquinone methyltransferase	1.79 ± 0.00	1.60 ± 0.33
Acetate production and utilization			
*ackA*	Acetate kinase	1.59 ± 0.05	0.82 ± 0.08
*acs **	Acetyl-CoA synthetase (AMP-forming)	3.99 ± 0.25	2.21 ± 0.35
*ldhA*	D-lactate dehydrogenase	3.03 ± 0.16	2.22 ± 0.15
*pflD*	Putative formate acetyltransferase 2	3.52 ± 0.24	2.76 ± 0.39
*pflB **	Pyruvate formate-lyase (inactive)	1.41 ± 0.08	1.42 ± 0.24
*poxB*	Pyruvate oxidase	3.51 ± 0.58	2.75 ± 0.22
Sigma factors			
*rpoA*	RNA polymerase subunit α	0.91 ± 0.13	0.69 ± 0.09
*rpoC*	RNA polymerase subunit β′	2.13 ± 0.04	1.01 ± 0.18
*rpoZ*	RNA polymerase subunit ω	2.70 ± 0.00	2.02 ± 0.46
*rpoD*	RNA polymerase sigma factor RpoD (σ^70^)	5.41 ± 0.40	9.87 ± 2.09
*rpoE **	RNA polymerase sigma factor RpoE (σ^24^)	1.77 ± 0.23	1.27 ± 0.07
*rpoH **	RNA polymerase sigma factor RpoH (σ^32^)	4.21 ± 0.49	2.42 ± 0.44
*rpoN*	RNA polymerase sigma factor RpoN (σ^54)^	4.84 ± 0.26	3.75 ± 0.92
*rpoS **	RNA polymerase sigma factor RpoS (σ^38^)	5.74 ± 0.81	1.92 ± 0.01
Regulators			
*arcA*	DNA-binding transcriptional dual regulator ArcA	1.96 ± 0.09	0.88 ± 0.04
*arcB*	Sensor histidine kinase ArcB	3.10 ± 0.03	2.00 ± 0.25
*Cra*	DNA-binding transcriptional dual regulator Cra	3.25 ± 0.05	1.82 ± 0.04
*cyaA **	Adenylate cyclase	2.47 ± 0.03	1.55 ± 0.03
*glcC **	DNA-binding transcriptional dual regulator GlcC	3.26 ± 0.26	2.30 ± 0.51
*iclR*	DNA-binding transcriptional repressor IcIR	2.50 ± 0.33	1.18 ± 0.29
*ihfA*	Integration host factor subunit α	1.33 ± 0.05	1.38 ± 0.04
*fadR*	DNA-binding transcriptional dual regulator FadR	3.95 ± 0.20	2.22 ± 0.04
*Fnr*	DNA-binding transcriptional dual regulator FNR	3.22 ± 0.18	2.81 ± 0.36
*narL*	DNA-binding transcriptional dual regulator NarL	1.10 ± 0.20	1.02 ± 0.29
Regulators to stress response			
*Rsd*	Regulator of sigma D	8.42 ± 0.33	18.1 ± 2.19
*soxR*	DNA-binding transcriptional dual regulator SoxR	30.6 ± 4.20	107.3 ± 10.3
*soxS*	DNA-binding transcriptional dual regulator SoxS	20.1 ± 0.55	81.8 ± 0.97
*spot*	Bifunctional (p)ppGpp synthase/hydrolase SpoT	75.1 ± 2.74	268.6 ± 30.2

^1^ According to the EcoCyc database [4]. * Indicates genes with transcriptional activation by CRP-cAMP according to the EcoCyc database.

## Data Availability

Data are contained within the article.

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
