# Peer review of "Transcriptional and Metabolic Response of a Strain of Escherichia coli PTS to a Perturbation of the Energetic Level by Modification of [ATP]/[ADP] Ratio"

_biotech, 2024, doi:10.3390/biotech13020010_

Round 1

Reviewer 1 Report

Comments and Suggestions for Authors

The paper provides valuable insights into the intricate relationship between the intracellular [ATP]/[ADP] ratio and Escherichia coli's cellular functions. The discussion on the overexpression of F1-ATPase genes in E. coli and its implications on glucose consumption, energy levels, and transcriptional responses in central carbon metabolism genes is well-articulated. However, there are areas where the review could be improved for better clarity and depth of analysis.

Providing more details on the experimental techniques employed, such as the methodology for uncoupling the cytoplasmic activity of the F1 subunit, would enhance the understanding of the study's approach.

conclusion is very lengthy. It should be based on the results obtained only.

providing insights into the mechanisms underlying the observed reduction in growth despite increased glucose transport and the implications of heightened stress response capabilities on cellular function would add depth to the analysis.

Expanding on the specific genes that were upregulated or downregulated in response to the ATP disruption and their functional significance would provide a more comprehensive understanding of the molecular mechanisms at play.

Further discussion on the broader implications of these findings in the context of bacterial physiology and adaptation would enhance the significance of the study.

In summary, while the manuscript offers valuable insights into the impact of perturbations in the [ATP]/[ADP] ratio on E. coli physiology and metabolism, further elaboration on experimental methods, physiological responses, gene expression data, and broader implications would strengthen the analysis and enrich the discussion.

Author Response

Dear Reviewer, 

Thanks very much for your kind suggestions and concerns on the first version of our submitted contribution. Here you can find the reply point by point to all your comments:

The paper provides valuable insights into the intricate relationship between the intracellular [ATP]/[ADP] ratio and Escherichia coli's cellular functions. The discussion on the overexpression of F1-ATPase genes in E. coli and its implications on glucose consumption, energy levels, and transcriptional responses in central carbon metabolism genes is well-articulated. However, there are areas where the review could be improved for better clarity and depth of analysis.

Providing more details on the experimental techniques employed, such as the methodology for uncoupling the cytoplasmic activity of the F1 subunit, would enhance the understanding of the study's approach.
Reply: A brief description of the uncoupling of the F1 cytoplasmic activity was included in the new lines 172-177, improving the understanding of the uncoupling ATPase activity. Additionally, a brief explanation of the uncoupled F1 ATPase activity was included in lines 269-271 in the Results section. 

conclusion is very lengthy. It should be based on the results obtained only. 
Reply: The new conclusion section was reduced, resulting in a concrete description of this contribution's main findings. The previous section was 444 words long, and the new section contains 353 words. Additionally, it is based only on obtained results. New lines 717-744.

providing insights into the mechanisms underlying the observed reduction in growth despite increased glucose transport and the implications of heightened stress response capabilities on cellular function would add depth to the analysis.
Reply: Dear reviewer, in the previous version of this contribution, we highlighted several mechanisms underlying the observed reduction in growth despite the reduced glucose transport associated with a possible increase in the carbon flux through the central carbon metabolism. In this new version, we highlighted this discussion in lines 397-399 and 417-435. Additionally, we moved the discussion on the stress response of the apt mutant to the new section: 4.8 Expression of stress response genes (lines 664-671). We consider that these modifications in the Discussion section provide a comprehensive discussion of the response of the apt mutant to the stress conditions resulting from the disruption of the [ATP]/[ADP] ratio.

Expanding on the specific genes that were upregulated or downregulated in response to the ATP disruption and their functional significance would provide a more comprehensive understanding of the molecular mechanisms at play.
Reply: Dear reviewer, Thanks for this suggestion. Nevertheless, we consider that the extent of the discussion on the physiological significance of the unregulated and downregulated genes in response to the [ATP]/[ADP] disruption in the PTS mutant of E. coli provides a comprehensive understanding of the physiological mechanisms involved in the response of the atp mutant, supported by previously results and observations reported by diverse authors. 
To provide more profound information on the fine molecular and physiological mechanisms developed by the ATP mutant in response to the ATP disturbance, we must develop new experimental strategies, such as analyzing metabolic fluxes and possibly conducting a proteomic study to correlate our proposed findings with the transcriptomic analysis. My research group is now considering these future approaches. 

Further discussion on the broader implications of these findings in the context of bacterial physiology and adaptation would enhance the significance of the study.
Reply: Dear Reviewer, Thanks for this suggestion. We include a new subsection: 4.9 Physiological significance of [ATP]/[ADP] ratio disruption in E. coli PTS- (new lines 703-715), describing the physiological implication of the AT wasting by cloning the atpAGD operon in E. coli and the relevance of the physiological response of our mutant for metabolic engineering purposes. 

In summary, while the manuscript offers valuable insights into the impact of perturbations in the [ATP]/[ADP] ratio on E. coli physiology and metabolism, further elaboration on experimental methods, physiological responses, gene expression data, and broader implications would strengthen the analysis and enrich the discussion.
Reply. Thanks for these proposed perspectives. As stated above, we are planning future experiments to characterize this mutant in depth, which we hope to submit for publication in the near future. 

Reviewer 2 Report

Comments and Suggestions for Authors

This study investigates the impact of disrupting the intracellular ATP/ADP ratio in E. coli through the overexpression of F1-ATPase genes. By uncoupling the cytoplasmic activity of the F1 subunit of ATP synthase in a PTS- mutant strain, termed PB12AGD+, the physiological and metabolic responses were analyzed. The mutant exhibited reduced growth despite increased glucose transport, heightened stress response capabilities, and redirected carbon flux towards pathways such as the pentose phosphate and Entner-Duodoroff pathways, along with increased respiratory capabilities and acetate production/utilization. Overall, this study is well-structured and methodologically sound. However, there are a couple of minor concerns that should be addressed:

(1) In lines 200-201, the statement "Acetic acid production is shown in Figure 1, panel B", refers to a part of the figure that is missing. Could the authors provide the data?

(2) For Figure 1,  please adjust the interval of the Y-axis to a smaller scale and represent the mean with standard deviation, which would improve the interpretation of the data.

(3) In Table 2, it is recommended that the authors include the P value and specify the statistical methodology employed to provide a comprehensive understanding of the data analysis process.

Author Response

Dear Reviewer, thanks for all your suggestions and concerns on the first version of our contribution. Here, you can find a reply to all your comments point by point. On behalf of the authors of this contribution, I hope to find a suitable new version.

This study investigates the impact of disrupting the intracellular ATP/ADP ratio in E. coli through the overexpression of F1-ATPase genes. By uncoupling the cytoplasmic activity of the F1 subunit of ATP synthase in a PTS- mutant strain, termed PB12AGD+, the physiological and metabolic responses were analyzed. The mutant exhibited reduced growth despite increased glucose transport, heightened stress response capabilities, and redirected carbon flux towards pathways such as the pentose phosphate and Entner-Duodoroff pathways, along with increased respiratory capabilities and acetate production/utilization. Overall, this study is well-structured and methodologically sound. However, there are a couple of minor concerns that should be addressed:

(1) In lines 200-201, the statement "Acetic acid production is shown in Figure 1, panel B", refers to a part of the figure that is missing. Could the authors provide the data?

Reply. Dear Reviewer, Thanks for your observation. The description in the referred lines corresponds to a former version of this manuscript. Finally, we excluded the acetic acid production profiles from Figure 1. The acetic acid production values were presented in the original subsection: 3.1. Physiological parameters of analyzed strains of E. coli in bioreactor cultures (now in the new lines: 237-240).

(2) For Figure 1,  please adjust the interval of the Y-axis to a smaller scale and represent the mean with standard deviation, which would improve the interpretation of the data.

Reply: Figure 1 was modified as suggested and inserted in the new document. As you can see, the SD values are visible for several data sets, as differences between replicates were smaller. We hope you find this new figure suitable.

(3) In Table 2, it is recommended that the authors include the P value and specify the statistical methodology employed to provide a comprehensive understanding of the data analysis process.

Reply. Dear Reviewer, we included in foot table 1 a brief description of the statistical procedures performed for data shown in this table as reported previously by my group: Rodriguez et al. Microbial Cell Factories 2013, 12:86. http://www.microbialcellfactories.com/content/12/1/86. New lines 264-267.